# Composable Coresets for Determinant Maximization: Greedy is Almost Optimal*

**Siddharth Gollapudi**
Microsoft Research
sgollapu@berkeley.edu

**Sepideh Mahabadi**
Microsoft Research
smahabadi@microsoft.com

**Varun Sivashankar**
Microsoft Research
varunsiva@princeton.edu

## Abstract

Given a set of $n$ vectors in $\mathbb{R}^d$, the goal of the *determinant maximization* problem is to pick $k$ vectors with the maximum volume. Determinant maximization is the MAP-inference task for determinantal point processes (DPP) and has recently received considerable attention for modeling diversity. As most applications for the problem use large amounts of data, this problem has been studied in the relevant *composable coreset* setting. In particular, [IMGR20] showed that one can get composable coresets with optimal approximation factor of $\tilde{O}(k)^k$ for the problem, and that a local search algorithm achieves an almost optimal approximation guarantee of $O(k)^{2k}$. In this work, we show that the widely-used Greedy algorithm also provides composable coresets with an almost optimal approximation factor of $O(k)^{3k}$, which improves over the previously known guarantee of $C^{k^2}$, and supports the prior experimental results showing the practicality of the greedy algorithm as a coreset. Our main result follows by showing a local optimality property for Greedy: swapping a single point from the greedy solution with a vector that was not picked by the greedy algorithm can increase the volume by a factor of at most $(1 + \sqrt{k})$. This is tight up to the additive constant 1. Finally, our experiments show that the local optimality of the greedy algorithm is even lower than the theoretical bound on real data sets.

## 1 Introduction

In the *determinant maximization* problem, we are given a set $P$ of $n$ vectors in $\mathbb{R}^d$, and a parameter $k \leq d$. The objective is to find a subset $S = \{v_1, \ldots, v_k\} \subseteq P$ consisting of $k$ vectors such that that the volume squared of the parallelepiped spanned by the points in the subset $S$ is maximized. Equivalently, the volume squared of a set $S$, denoted by vol$(S)$, is equal to the determinant of the Gram matrix of the vectors in $S$. Determinant maximization is the MAP-inference of determinantal point processes, and both of these problems as well as their variants have found numerous applications in data summarization, machine learning, experimental design, and computational geometry. In particular, the determinant of a subset of points is one way to measure the *diversity* of the subset, and thus they have been studied extensively over the last decade in this context [MJK17, GCGS14, KT+12, CGGS15, KT11, YFZ+16, LCYO16].

The best approximation factor for the problem in this regime is due to the work of [Nik15] who shows a factor of $e^k$, and it is known that an exponential dependence on $k$ is necessary [CMI13] unless P = NP. However, the most common algorithm used for this problem in practical applications is a natural *greedy* algorithm. In this setting, the algorithm first picks the vector with the largest norm,

---

*This is an equal contribution paper

and then greedily picks the vector with largest perpendicular component to the subspace spanned by the current set of picked vectors, thus maximizing the volume greedily in each iteration. This algorithm is known to have an approximation factor of $(k!)^2$[ÇMI09].

As in most applications of determinant maximization one needs to work with large amounts of data, there has been an increasing interest in studying determinant maximization in large data models of computation [MJK17, WIB14, PJG$^+$14, MKSK13, MKBK15, MZ15, BENW15]. One such model that we focus on in this work is the *composable coreset* setting [IMMM14]. Intuitively, composable coresets are small "summaries" of a data set with the composability property: for the summaries of multiple datasets, the union of the summaries should make a good summary for the union of the datasets. More precisely, in this setting, instead of a single set of vectors $P$, there are $m$ sets $P_1, \ldots, P_m \subseteq \mathbb{R}^d$. In this context, a mapping function $c$ that maps a point set to one of its subsets is called $\alpha$-*composable coreset* for determinant maximization, if for any collection of point sets $P_1, \ldots, P_m$,

$$\mathsf{MAXDET}_k \left( \cup_{i=1}^m c(P_i) \right) \geq \frac{1}{\alpha} \cdot \mathsf{MAXDET}_k \left( \cup_{i=1}^m P_i \right) \tag{1}$$

where $\mathsf{MAXDET}_k$ is used to denote the maximum achievable determinant with parameter $k$. (Similarly, $\mathsf{MAXVOL}_k$ is used to denote the maximum volume, with $\mathsf{MAXVOL}_k^2 = \mathsf{MAXDET}_k$.) For clarity, we note that the mapping function $c$ can only view its input data set $P_i$ and has no knowledge of other data sets while constructing $c(P_i)$. [IMMM14] showed that a composable coreset for a task automatically gives an efficient distributed and an efficient streaming algorithm for the same task.

Indeed, composable coresets have been used for determinant maximization. In particular, [IMGR20, MIGR19], presented a composable coreset of size $O(k \log k)$ with approximation factor of $\tilde{O}(k)^k$ using spectral spanners, which they showed to be almost tight. In particular, the best approximation factor one can get is $\Omega(k^{k-o(k)})$ (Theorem 1.4). As the above algorithm is LP-based and does not provide the best performance in practice, they proposed to use the greedy algorithm followed by a local search procedure, and showed that this simple algorithm also yields a coreset of size $k$ with an almost optimal approximation guarantee of $O(k)^{2k}$. They also proved that the greedy algorithm alone yields a $C^{k^2}$ guarantee for composable coresets, which is far larger than the optimal approximation of $\tilde{O}(k)^k$ for this problem.

Since the greedy algorithm provides a very good performance in practice [MIGR19, MKSK13], an improved analysis of the greedy algorithm in the coreset setting is very desirable. Furthermore, both of these prior work implied that greedy performs well in practice in the context of distributed and composable coreset settings [MKSK13], and in particular its performance is comparable to that of the local search algorithm for the problem [MIGR19].

**Our contribution.** In this paper, we close this theoretical gap: we prove that the greedy algorithm provides a $O(k)^{3k}$-composable coreset for the determinant maximization problem (Theorem 4). This explains the very good performance of this algorithm on real data previously shown in [MIGR19, MKSK13]. We achieve this by proving an elegant linear algebra result on the local optimality of the greedy algorithm: swapping a single point from the greedy solution with a vector that was not picked by the greedy algorithm can increase the volume by a factor of at most $(1 + \sqrt{k})$. We further show that this is tight up to the additive constant 1. As an application of our result, we give a proof that the locality property can recover and in fact marginally improve the $k!$ guarantee of the greedy algorithm of [ÇMI09] for the offline volume maximization problem.

Finally, in Section 4, we run experiments to measure the local optimality of the greedy algorithm on real data, and show that this number is much smaller in practice than the worst case theoretically guaranteed bound. In fact, in our experiments this number is always less than 1.5 even for $k$ even as large as 300. Again this explains the practical efficiency of the greedy algorithm as a coreset shown in [MIGR19, MKSK13].

## 1.1 Preliminaries

### 1.1.1 The Greedy Algorithm

Recall the standard offline setting for determinant maximization, where one is required to pick $k$ vectors out of the $n$ vectors in $P$ of maximum volume. Here, [ÇMI09] showed that greedily picking

the vector with the largest perpendicular distance to the subspace spanned by the current solution (or equivalently, greedily picking the vector that maximizes the volume as in Algorithm 1) outputs a set of vectors that is within $k!$ of the optimal volume. Formally, if $\mathsf{Greedy}(P)$ is the output of Algorithm 1, then

$$\mathsf{vol}(\mathsf{Greedy}(P)) \geq \frac{\mathsf{MAXVOL}_k(P)}{k!} \tag{2}$$

### 1.1.2 Local Search for Composable Coresets

In [MIGR19], the authors show that the greedy algorithm followed by the local search procedure with parameter $\epsilon$ (as described in Algorithm 2) provides a $(2k(1+\epsilon))^{2k}$-composable coreset for determinant maximization. A locally optimal solution can thus be naturally defined as follows:

**Definition 1** $((1+\epsilon)$-Locally Optimal Solution). *Given a point set $P \subseteq \mathbb{R}^d$ and $c(P) \subseteq P$ with $|c(P)| = k$, we say $c(P)$ is a $(1+\epsilon)$-locally optimal solution for volume maximization if for any $v \in c(P)$ and any $w \in P \setminus c(P)$,*

$$\mathsf{vol}(c(P) - v + w) \leq (1+\epsilon)\,\mathsf{vol}(c(P)) \tag{3}$$

Given the output of the greedy algorithm $\mathsf{Greedy}(P)$, one can obtain a locally optimal solution using a series of swaps: if the volume of the solution can be increased by a factor of $(1+\epsilon)$ by swapping a vector in the current solution with a vector in the point set $P$ that has not been included, we make the swap. Since $\mathsf{vol}(\mathsf{Greedy}(P))$ is within a factor of $k!$ of the optimal, we will make at most $\frac{k \log k}{\log(1+\epsilon)}$ swaps. This is precisely the local search algorithm (Algorithm 2). For any point set $P$, we denote the output of Algorithm 2 by $\mathsf{LS}(P)$.

In [MIGR19], the authors prove that local search yields a $O(k)^{2k}$-composable coreset for determinant maximization. Formally, they prove the following.

**Theorem 2.** *Let $P_1, \ldots, P_m \subseteq \mathbb{R}^d$. For each $i = 1, \ldots, m$, let $\mathsf{LS}(P_i)$ be the output of the local search algorithm (Algorithm 2) with parameter $\epsilon$. Then*

$$\mathsf{MAXDET}_k(\cup_{i=1}^m P_i) \leq (2k(1+\epsilon))^{2k}\,\mathsf{MAXDET}_k(\cup_{i=1}^m \mathsf{LS}(P_i)) \tag{4}$$

**Remark 3.** *Even though [MIGR19] treats $\epsilon$ as a small constant in $[0, 1]$, the proof for Theorem 2 above holds for any non-negative $\epsilon$.*

## 1.2 Outline of our approach

In [MIGR19], the authors prove Theorem 2 for local search using a reduction to a related problem called $k$-directional height. The authors then use similar ideas to prove that the output of the greedy algorithm is also a composable coreset for determinant maximization. However, since we do not know a priori whether greedy is $(1+\epsilon)$-locally optimal, the guarantee they obtain is significantly weaker: they only prove that the greedy algorithm yields a $((2k) \cdot 3^k)^{2k} = C^{k^2}$-composable coreset for determinant maximization. This is clearly far from the desired bound of $k^{O(k)}$.

To improve the analysis of the greedy algorithm in the coreset setting, we ask the following natural question:

*Can we prove that the output of the greedy algorithm is already locally optimal?*

We answer this question positively. Our main result is Theorem 5, where we show that for any point set $P$, $\mathsf{Greedy}(P)$ is a $(1+\sqrt{k})$-locally optimal solution. In other words, the greedy algorithm has the same guarantee as local search with the parameter $\epsilon = \sqrt{k}$. This circumvents the loose reduction from greedy to the $k$-directional height problem and directly implies the following improved guarantee for the greedy algorithm in the coreset setting:

**Theorem 4.**

$$\mathsf{MAXDET}_k(\cup_{i=1}^m P_i) \leq (2k(1+\sqrt{k}))^{2k}\,\mathsf{MAXDET}_k(\cup_{i=1}^m \mathsf{Greedy}(P_i)) \tag{5}$$

Thus, the greedy algorithm also provides a $(2k(1+\sqrt{k}))^{2k} = k^{O(k)}$-composable coreset for determinant maximization, which is near the optimal $\Omega(k^{k-o(k)})$.

Section 2 is dedicated to proving that greedy is $(1 + \sqrt{k})$-locally optimal (Theorem 5). We also show that this local optimality result of $(1 + \sqrt{k})$ for the greedy algorithm is tight up to the additive constant 1. In Section 4 we show that on real and random datasets, the local optimality constant $\epsilon$ is much smaller than the bound of $1 + \sqrt{k}$, which serves as an empirical explanation for why greedy performs much better in practice than what the theoretical analysis suggests.

---

**Algorithm 1** Greedy Algorithm

---

**Input:** A point set $P \subset \mathbb{R}^d$ and integer $k$.
**Output:** A set $\mathcal{C} \subset P$ of size $k$.
Initialize $\mathcal{C} = \emptyset$.
**for** $i = 1$ **to** $k$ **do**
    Add $\operatorname{argmax}_{p \in P \setminus \mathcal{C}} \operatorname{vol}(\mathcal{C} + p)$ to $\mathcal{C}$.
**end for**
**Return** $\mathcal{C}$.

---

**Algorithm 2** Local Search Algorithm

---

**Input:** A point set $P \subset \mathbb{R}^d$, integer $k$, and $\epsilon > 0$.
**Output:** A set $\mathcal{C} \subset P$ of size $k$.
Initialize $\mathcal{C} = \emptyset$.
**for** $i = 1$ **to** $k$ **do**
    Add $\operatorname{argmax}_{p \in P \setminus \mathcal{C}} \operatorname{vol}(\mathcal{C} + p)$ to $\mathcal{C}$.
**end for**
**repeat**
    If there are points $q \in P \setminus \mathcal{C}$ and $p \in \mathcal{C}$ such that

$$\operatorname{vol}(\mathcal{C} + q - p) \geq (1 + \epsilon)\operatorname{vol}(\mathcal{C})$$

    replace $p$ with $q$.
**until** No such pair exists.
**Return** $\mathcal{C}$.

---

## 2 Greedy is Locally Optimal

**Theorem 5** (Local Optimality). *Let $V := \mathsf{Greedy}(P) = \{v_1, \ldots, v_k\} \subseteq P$ be the output of the greedy algorithm. Let $v_{k+1} \in P \setminus V$ be a vector not chosen by the greedy algorithm. Then for all $i = 1, \ldots, k$,*

$$\operatorname{vol}(V - v_i + v_{k+1}) \leq (1 + \sqrt{k})\operatorname{vol}(V) \tag{6}$$

*Proof.* If $\operatorname{rank}(P) < k$, then the result is trivial. So we may assume $\operatorname{rank}(P) \geq k$ and $V$ is linearly independent. Fix any $v_i \in V$. Our goal is to show that $\operatorname{vol}(V - v_i + v_{k+1}) \leq (1 + \sqrt{k})\operatorname{vol}(V)$. This trivially holds when $i = k$ by the property of the greedy algorithm, so assume $1 \leq i \leq k - 1$.

Let $\{v'_1, \ldots, v'_k, v'_{k+1}\}$ be the set of orthogonal vectors constructed by performing the Gram-Schmidt algorithm on $\{v_1, \ldots, v_k, v_{k+1}\}$. Formally, let $\mathcal{G}_t = \operatorname{span}\{v_1, \ldots, v_t\}$. Define $v'_1 = v_1$ and $v'_t = v_t - \Pi(\mathcal{G}_{t-1})(v_t)$ for $t = 2, \ldots, k, k+1$, where $\Pi(\mathcal{G})(v)$ denotes the projection of the vector $v$ onto the subspace $\mathcal{G}$. Note that

$$\operatorname{vol}(V) = \prod_{j=1}^{k} \|v'_j\|_2$$

For each $j = i+1, \ldots, k, k+1$, write

$$v_j = \Pi(\mathcal{G}_{i-1})(v_j) + \sum_{l=i}^{j} \alpha_l^j v'_l$$
$$:= \Pi(\mathcal{G}_{i-1})(v_j) + w_j$$

We must have that $|\alpha_j^l| \leq 1$ by the greedy algorithm because if $|\alpha_l^j| > 1$, the vector $v_j$ would have been chosen before $v_l$. Further, $\alpha_j^j = 1$ by definition of Gram-Schmidt. The vector $w_j$ is what remains of $v_j$ once we subtract its projection onto the first $i - 1$ vectors.

We are interested in bounding the following quantity:

$$\text{vol}(V - v_i + v_{k+1}) = \text{vol}(v_1, \ldots, v_{i-1}, v_{i+1}, \ldots, v_k, v_{k+1})$$
$$= \text{vol}(v'_1, \ldots, v'_{i-1}, v_{i+1}, \ldots, v_k, v_{k+1})$$
$$= \text{vol}(v'_1, \ldots, v'_{i-1}, w_{i+1}, \ldots, w_k, w_{k+1})$$
$$= \text{vol}(v'_1, \ldots, v'_{i-1}) \cdot \text{vol}(w_{i+1}, \ldots, w_k, w_{k+1})$$
$$= \left( \prod_{j=1}^{i-1} \|v'_j\|_2 \right) \cdot \text{vol}(w_{i+1}, \ldots, w_k, w_{k+1})$$

Therefore, it suffices to prove the following:

$$\text{vol}(w_{i+1}, \ldots, w_k, w_{k+1}) \le (1 + \sqrt{k}) \prod_{j=i}^{k} \|v'_j\|_2 \tag{7}$$

To establish this, we consider two cases. Recall that $v'_{k+1} = v_{k+1} - \Pi(\mathcal{G}_k)(v_k)$. We analyze the cases where $v'_{k+1} \ne 0$ and $v'_{k+1} = 0$ separately, although the ideas are similar. In Claim 7 and Claim 8 below, we establish the desired bound stated in Eq. (7) for $v'_{k+1} \ne 0$ and $v'_{k+1} = 0$ respectively. Theorem 5 then follows immediately. $\square$

To prove Claim 7 and Claim 8, the following well-known lemma will be useful. A proof can be found in [DZ07].

**Lemma 6** (Matrix Determinant Lemma). *Suppose $M$ is an invertible matrix. Then*

$$\det(M + uv^T) = (1 + v^T M^{-1} u) \det(M) \tag{8}$$

**Claim 7.** *Suppose $v'_{k+1} \ne 0$. Then*

$$\text{vol}(w_{i+1}, \ldots, w_k, w_{k+1}) \le \left( \sqrt{k+1} \right) \prod_{j=i}^{k} \|v'_j\|_2 \tag{9}$$

*Proof.* Define the matrix $B = [w_{i+1}| \cdots |w_k|w_{k+1}]$. Note that $\det(B^T B) = \text{vol}(w_{i+1}, \ldots, w_k, w_{k+1})^2$ is the quantity we are interested in bounding. For clarity,

$$B^T = \begin{bmatrix} \alpha_i^{i+1} v'_i + v'_{i+1} \\ \alpha_i^{i+2} v'_i + \alpha_{i+1}^{i+2} v'_{i+1} + v'_{i+2} \\ \vdots \\ \alpha_i^{k+1} v'_i + \cdots + \alpha_k^{k+1} v'_k + v'_{k+1} \end{bmatrix}$$

We define the matrix $A$ by just removing the $v'_i$ terms from $B$ as follows:

$$A^T = \begin{bmatrix} v'_{i+1} \\ \alpha_{i+1}^{i+2} v'_{i+1} + v'_{i+2} \\ \vdots \\ \alpha_{i+1}^{k+1} v'_{i+1} + \cdots + \alpha_k^{k+1} v'_k + v'_{k+1} \end{bmatrix}$$

Since $\langle v'_i, v'_j \rangle = 0$ for all $j \ne i$, we have that

$$B^T B = A^T A + uu^T$$

where the column vector $u$ is given by

$$u = \|v'_i\| \cdot \begin{pmatrix} \alpha_i^{i+1} & \alpha_i^{i+2} & \cdots & \alpha_i^{k+1} \end{pmatrix}$$

Since $|\alpha_i^j| \le 1$ for $j = i+1, \ldots, k+1$ by the nature of the greedy algorithm, we have that

$$\|u\|_2^2 \le (k - i + 1) \|v'_i\|_2^2 \le k \|v'_i\|_2^2 \tag{10}$$

We now bound the desired volume quantity. Let $M = A^T A$. $M$ is clearly a positive semi-definite matrix. In fact, because we assumed that $v'_{k+1} \neq 0$, it will turn out that $M$ is positive definite and thus invertible. For now, assume that $M^{-1}$ exists. We will compute the inverse explicitly later.

$$
\begin{aligned}
\mathsf{vol}(w_{i+1}, \ldots, w_k, w_{k+1})^2 &= \det(B^T B) \\
&= \det(A^T A + u u^T) \\
&= (1 + u^T M^{-1} u) \det(M) \qquad \text{[by Lemma 6]} \\
&\leq \left(1 + k\|v'_i\|^2 \lambda_{\max}(M^{-1})\right) \det(M)
\end{aligned}
$$

where $\lambda_{\max}(M^{-1})$ is the largest eigenvalue of $M^{-1}$. We will now show that $M^{-1}$ does in fact exist and bound $\lambda_{\max}(M^{-1})$. Consider the matrix $E$ and $W$ defined as follows:

$$
E = \begin{bmatrix} 1 & 0 & & \cdots & 0 \\ \alpha_{i+1}^{i+2} & 1 & 0 & \cdots & 0 \\ \vdots & & & & \\ \alpha_{i+1}^{k+1} & & \cdots & \alpha_k^{k+1} & 1 \end{bmatrix}
\qquad
W^T = \begin{bmatrix} v'_{i+1} \\ v'_{i+2} \\ \vdots \\ v'_{k+1} \end{bmatrix}
$$

It is easy to check that $E W^T = A^T$. Therefore,

$$
M = A^T A = E W^T W E^T = E D E^T
$$

where $D$ is the diagonal matrix given by

$$
D = \mathrm{diag}\left(\|v'_{i+1}\|_2^2, \ldots, \|v'_{k+1}\|_2^2\right)
$$

It is easy to see that $E$ has all eigenvalues equal to 1, and so must be invertible with determinant 1. The same is obviously true for $E^T$, $E^{-1}$ and $(E^T)^{-1}$ as well. It follows that

$$
\det(M) = \det(D) = \|v'_{i+1}\|^2 \cdots \|v'_{k+1}\|^2
$$

Since $v'_{k+1} \neq 0$, we have that $\|v'_j\|_2 > 0$ for all $j$, so $D^{-1}$ clearly exists. It follows that

$$
M^{-1} = (A^T A)^{-1} = (E^T)^{-1} D^{-1} E^{-1}
$$

$$
\lambda_{\max}(M^{-1}) \leq \lambda_{\max}(D^{-1}) = \frac{1}{\|v'_{k+1}\|^2}
$$

Therefore,

$$
\begin{aligned}
\mathsf{vol}(w_{i+1}, \ldots, w_k, w_{k+1})^2 &\leq (1 + k\|v'_i\|^2 \lambda_{\max}(M^{-1})) \det(M) \\
&\leq \left(1 + \frac{k\|v'_i\|^2}{\|v'_{k+1}\|^2}\right) \det(M) \\
&= \det(M) + k \prod_{j=i}^{k} \|v'_j\|_2^2 \\
&\leq (1 + k) \prod_{j=i}^{k} \|v'_j\|_2^2
\end{aligned}
$$

$\square$

**Claim 8.** *Suppose $v'_{k+1} = 0$. Then*

$$
\mathsf{vol}(w_{i+1}, \ldots, w_k, w_{k+1}) \leq \left(1 + \sqrt{k}\right) \prod_{j=i}^{k} \|v'_j\|_2 \tag{11}
$$

*Proof.* The idea for this proof is similar to the previous claim. However, the main catch is that decomposing $B^T B$ into $A^T A + u u^T$ (as defined in the proof of Claim 7) is no longer helpful because $v'_{k+1} = 0$ implies that $A^T A$ is not invertible. However, there is a simple workaround.

Define the matrix $B' = [w_{k+1}|w_{i+1}|\cdots|w_k]$. Note that $\det((B')^T B') = \mathrm{vol}(w_{i+1}, \ldots, w_k, w_{k+1})^2$ is the quantity we are interested in bounding. Recall that $v'_{k+1} = 0$ by assumption. For clarity,

$$(B')^T = \begin{bmatrix} \alpha_i^{k+1} v'_i + \cdots + \alpha_k^{k+1} v'_k \\ \alpha_i^{i+1} v'_i + v'_{i+1} \\ \alpha_i^{i+2} v'_i + \alpha_{i+1}^{i+2} v'_{i+1} + v'_{i+2} \\ \vdots \\ \alpha_i^k v'_i + \cdots + v'_k \end{bmatrix}$$

Note that $(B')^T$ is the same as $B^T$ from the proof of Claim 7 except for moving the last row to the position of the first row. This change is just for convenience in this proof.

Define the following coefficients matrix $C \in \mathbb{R}^{(k-i+1)\times(k-i+1)}$:

$$C = \begin{bmatrix} \alpha_i^{k+1} & \cdots & & & \alpha_k^{k+1} \\ \alpha_i^{i+1} & 1 & 0 & 0 & 0 \\ \alpha_i^{i+2} & \alpha_{i+1}^{i+2} & 1 & 0 & 0 \\ \vdots & & & & \\ \alpha_i^k & \alpha_{i+1}^k & & \cdots & 1 \end{bmatrix} = \begin{bmatrix} 1 & 0 & & & 0 \\ \alpha_i^{i+1} & 1 & 0 & 0 & 0 \\ \alpha_i^{i+2} & \alpha_{i+1}^{i+2} & 1 & 0 & 0 \\ \vdots & & & & \\ \alpha_i^k & \alpha_{i+1}^k & & \cdots & 1 \end{bmatrix} + \begin{bmatrix} 1 \\ 0 \\ \vdots \\ \vdots \\ 0 \end{bmatrix} \begin{bmatrix} (\alpha_i^{k+1}-1) & \alpha_{i+1}^{k+1} & \cdots & \alpha_k^{k+1} \end{bmatrix}$$

$$:= C' + e_1 x^T$$

Define $W' = [v'_i|\cdots|v'_k]$. By construction, $(B')^T = C(W')^T$. Therefore

$$(W')^T W := D' = \mathrm{diag}\left(\|v'_i\|_2^2, \ldots, \|v'_k\|_2^2\right)$$

It follows that

$$\det((B')^T B') = \det(C(W')^T W' C^T)$$
$$= \det(C)^2 \det(D')$$
$$= \det(C)^2 \prod_{j=i}^k \|v'_j\|_2^2$$

It remains to show that $|\det(C)| \leq (1 + \sqrt{k})$. We may assume that $\alpha_i^{k+1} \geq 0$ by taking the negative of the first column if necessary. This does not affect the magnitude of the determinant. Note that all eigenvalues of $C'$ and $(C')^{-1}$ are 1. Further,

$$\|x\|_2^2 \leq k - i + 1 \leq k \tag{12}$$

$$|\det(C)| = |(1 + x^T(C')^{-1}e_1)| \cdot |\det(C')| \qquad \text{[by Lemma 6]}$$
$$= |1 + x^T(C')^{-1}e_1|$$
$$\leq 1 + \sqrt{k}\lambda_{\max}((C')^{-1})$$
$$= 1 + \sqrt{k}$$

$\square$

We now provide a simple example where the output of the greedy algorithm is at best $\sqrt{k}$-locally optimal, thus demonstrating that the locality result for greedy is optimal up to the constant 1.

**Theorem 9** (Tightness of Local Optimality). *There exists a point set $P = \{v_1, \ldots, v_k, v_{k+1}\}$ from which the greedy algorithm picks $V = \{v_1, \ldots, v_k\}$, and*

$$\frac{\mathrm{vol}(V - v_1 + v_{k+1})}{\mathrm{vol}(V)} = \sqrt{k} \tag{13}$$

*Proof.* Let $P = \{v_1, \ldots, v_k, v_{k+1}\}$ where $v_1 \in \mathbb{R}^k$ is the vector of all ones and $v_i = \sqrt{k} e_{i-1}$ for $i = 2, \ldots, k+1$. Since the magnitude of every vector in $P$ is $\sqrt{k}$, the greedy algorithm could start by picking $v_1$. The greedy algorithm will then pick any $k-1$ of the remaining $k$ vectors. Without loss in generality, assume that the algorithm picks $V = \{v_1, \ldots, v_k\}$. Then $\mathsf{vol}(V) = (\sqrt{k})^{k-1}$. On the other hand, $\mathsf{vol}(V - v_1 + v_{k+1}) = (\sqrt{k})^k$. The result follows. $\qquad\square$

## 3   Application to Standard Determinant Maximization

The greedy algorithm for volume maximization was shown to have an approximation factor of $k!$ in [ÇMI09]. We provide a completely new proof for this result with a slightly improved approximation factor.

**Theorem 10.** *Let $P$ be a point set, $\mathsf{Greedy}(P) = \{v_1, \ldots, v_k\}$ the output of the greedy algorithm, and $\mathsf{MAXVOL}_k(P)$ the maximum volume of any subset of $k$ vectors from $P$. Then*

$$\mathsf{vol}(\mathsf{Greedy}(P)) \geq \frac{\mathsf{MAXVOL}_k(P)}{\prod_{i=2}^{k}(1 + \sqrt{i})} \tag{14}$$

*Proof.* Let $S \subseteq P$ be the set of $k$ vectors with maximum volume. Without loss of generality and for simplicity of exposition, we assume that $\mathsf{Greedy}(P) \cap S = \varnothing$ (the proof still goes through if this is not the case). We will order $S$ in a convenient manner.

Consider the set $W_1 = \{v_1\} \cup S$ with $k+1$ elements. Perform the greedy algorithm on $W_1$ with $k$ steps. Clearly, greedy will choose $v_1$ first and then some $k-1$ of the remaining vectors. Label the left out vector $w_1$.

Inductively define $W_{i+1} = \{v_1, \ldots, v_i, v_{i+1}\} \cup (S - \{w_1, \ldots, w_i\})$, which has size $k+1$. Perform greedy on $W_{i+1}$ with $k$ steps. The first $i+1$ vectors chosen will be $v_1, \ldots, v_i, v_{i+1}$ by definition. Call the left out vector $w_{i+1}$. We now have an ordering for $S = \{w_1, \ldots, w_k\}$.

Starting with the greedy solution, we will now perform $k$ swaps to obtain the optimal solution. Each swap will increase the volume by a factor of at most $1 + \sqrt{k}$. Initially, our solution starts with $\mathsf{Greedy}(P) = \{v_1, \ldots, v_k\}$. Note that this is also the output of greedy when applied to the set $\mathsf{Greedy}(P) \cup \{w_k\} = W_k$. Swapping in $w_k$ in place of $v_k$ increases our volume by a factor of at most $1 + \sqrt{k}$.

Our current set of vectors is now $\{v_1, \ldots, v_{k-1}, w_k\}$. By the ordering on $S$, this is also the greedy output on the set $W_{k-1} = \{v_1, \ldots, v_{k-1}, w_{k-1}, w_k\}$. Therefore, we may swap in $w_{k-1}$ in place of $v_{k-1}$ in our current set of vectors by increasing the volume by at most a factor of $(1 + \sqrt{k})$. Proceeding in this manner, we can perform $k$ swaps to obtain the optimal solution from the greedy solution by increasing our volume by a factor of at most $(1 + \sqrt{k})^k$.

To obtain the slightly better approximation factor in the theorem statement, we observe that in the proof of Theorem 5, swapping out the $i^{\text{th}}$ vector from the greedy solution for a vector that was not chosen increases the volume only by a factor of $(1 + \sqrt{k+1-i}) \leq 1 + \sqrt{k}$ (Eq. (10),Eq. (12)), and that swapping out the $k^{\text{th}}$ vector does not increase the volume at all. Therefore, the approximation factor of greedy is at most

$$\prod_{i=1}^{k-1}(1 + \sqrt{k+1-i}) = \prod_{i=2}^{k}(1 + \sqrt{i})$$

$\qquad\square$

**Remark 11.** *Note that $\prod_{i=2}^{k}(1 + \sqrt{i}) < 2^k\sqrt{k!}$ for $k \geq 7$, which is $(k!)^{\frac{1}{2}+o(1)}$. While the improvement in the approximation factor is quite small, we emphasize that the proof idea is very different from the $k!$ guarantee obtained in [ÇMI09].*

# 4 Experiments

In this section, we measure the local optimality parameter for the greedy algorithm empirically. We use two real world datasets, both of which were used as benchmarks for determinant maximization in immediately related work ([MIGR19, LJS16]:

- **MNIST** [LBBH98], which has 60000 elements, each representing a 28-by-28 bitmap image of a hand-drawn digit;
- **GENES** [BQK⁺14], which has 10000 elements, with each representing the feature vector of a gene. The data set was initially used in identifying a diverse set of genes to predict breast cancer tumors. After removing the elements with some unknown values, we have around 8000 points.

We measure the local optimality parameter both as a function of $k$, and as a function of the data set size as explained in the next two subsections.

## 4.1 Local Optimality for Real and Random Datasets as a Function of $k$

**Experiment Setup:** For both MNIST and GENES, we consider a collection of $m = 10$ data sets, each with $n = 3000$ points chosen uniformly at random from the full dataset. We ran the greedy algorithm for $k$ from 1 to 20 and measured the local optimality value $(1 + \epsilon)$ as a fucntion of $k = 2, 4, \ldots, 20$ for each of the 10 data sets in the collection. More precisely, for each such $k$, we took the maximum value of $(1 + \epsilon)$ over every data set in the collection. The reason we take the worst value of $(1 + \epsilon)$, is that in the context of composable coresets, we require the guarantee to hold for each individual data set to be $(1 + \epsilon)$-locally optimal. We repeated this process for 5 iterations and took the average. We plot this value as a function of $k$.

Further, to compare against a random data set, for both MNIST and GENES, we repeated the above experiment against a set of random points of the same dimension sampled uniformly at random from the unit sphere.

**Results:** As shown in Fig. 1, while the real world data sets have local optimality value $(1 + \epsilon)$ higher than the random data sets, they are both significantly lower than (less than $1.4$) the theoretical bound of $(1 + \sqrt{k})$. This suggests that real world data sets behave much more nicely and are closer to random than the worst case analysis would suggest, which explains why greedy does so well in practice.

For the purpose of diversity maximization, the regime of interest is when $k \ll n$. However, we wanted to verify that the local optimality value does not increase much even when $k$ is much larger and closer to $n$. Since measuring local optimality is expensive when both $k$ and $n$ are large, we ran the same experiment again, except with $n = 300$ points per point set, and measuring the local optimality at $k = 1, 50, 100, \ldots, 300$ in steps of 50. Again, as seen in Fig. 2, local optimality stays much below $1 + \sqrt{k}$ (in fact less than $1.5$) for larger values of $k$ as well.

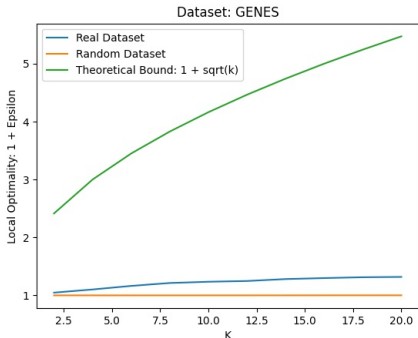 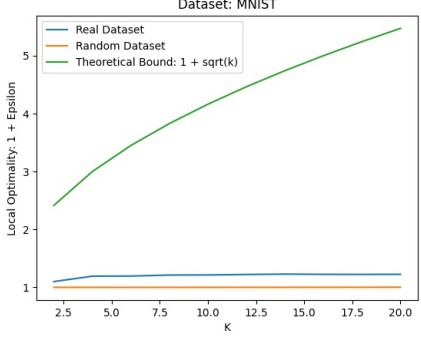

Figure 1: Local Optimality $(1 + \epsilon)$ against $k$ for GENES and MNIST datasets, and random datasets of the same dimension. Each stream had 10 point sets of size 3000, with $k$ ranging from 1 to 20.

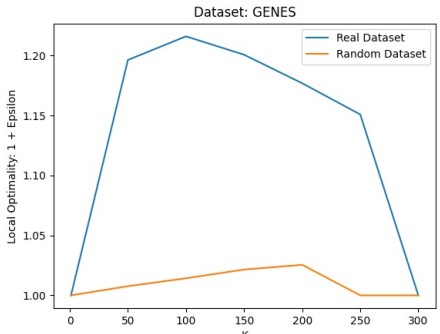
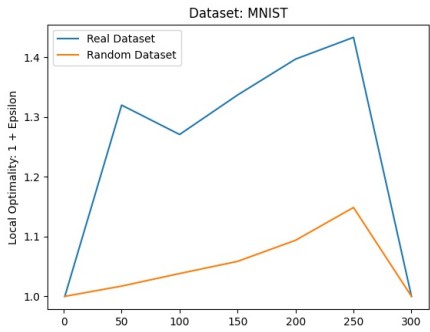

Figure 2: Local Optimality $(1 + \epsilon)$ against $k$ for GENES and MNIST datasets, and random datasets of the same dimension. Each stream had $10$ point sets of size $300$, with $k$ from $1$ to $300$ in steps of $50$. Note that when $k \in \{1, n\}$, we trivially have that $(1 + \epsilon) = 1$.

## 4.2 Local Optimality as a Function of the Size of Point Sets

**Experiment Setup:** Here, we fix the value of $k \in \{5, 10, 15, 20\}$ and compute the local optimality value $(1 + \epsilon)$ while increasing the size of the point sets. The point set size is varied from $500$ to $4000$ in intervals of $500$. For each point set size, we chose a stream of $10$ random point sets from the dataset and took the maximum value over $10$ iterations. Once again, we did this on MNIST and GENES and took the average of $5$ iterations.

**Results:** As shown in Fig. 3, the local optimality parameter remains very low (lower than $1.2$) regardless of the number of points in the data set, which is much smaller than $(1 + \sqrt{k})$.

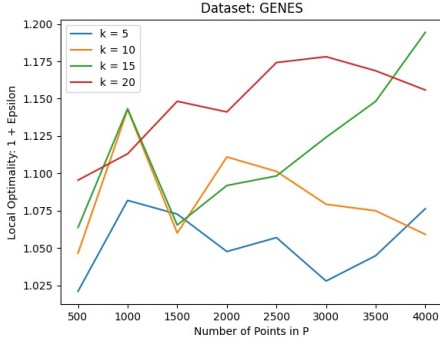
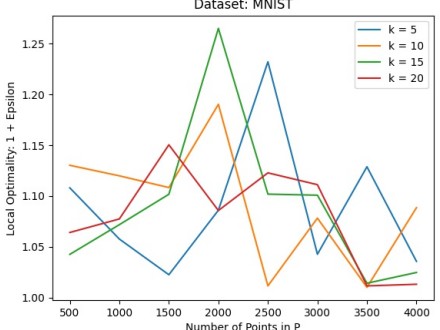

Figure 3: Local Optimality $(1 + \epsilon)$ against Number of Points in the Base Set for $k = 5, 10, 15, 20$.

## 5   Conclusion

In this work, we provided an almost tight analysis of the greedy algorithm for determinant maximization in the composable coreset setting: we improve upon the previous known bound of $C^{k^2}$ to $O(k)^{3k}$, which is optimal upto the factor $3$ in the exponent. We do this by proving a result on the local optimality of the greedy algorithm for volume maximization. We also support our theoretical analysis by measuring the local optimality of greedy over real world data sets. It remains an interesting question to tighten the constant in the exponent or otherwise provide a lower bound showing that $3$ is in fact optimal.

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
