# OpenReview forum: "Composable Coresets for Determinant Maximization: Greedy is Almost Optimal"
_NeurIPS.cc/2023/Conference — NeurIPS 2023 poster_

### Official Review · Reviewer_KpLA · 2023-06-23

**Soundness:** 3 good
**Presentation:** 3 good
**Contribution:** 3 good
**Rating:** 5
**Confidence:** 5

**Summary:**

The paper investigates the composable coresets for the determinant maximization problem that aims to pick $k$ vectors with the maximum volume. The authors prove that the widely-used greedy algorithm also provides composable coresets with an approximation factor of $O(k)^{3k}$, followed by showing a local optimality property for greedy. Empirical results show that the local optimality of the greedy algorithm is even lower.

**Strengths:**

- The topic of DPP is relevant.
- The technical result is solid.
- The observation of local optimality for greedy is of independent interest.

**Weaknesses:**

- The paper is a follow-up work of [IMGR20]. The improved analysis for the greedy algorithm is interesting but not the first result of a composable coreset.
- The studied problem is limited, e.g., only consider unconstrained DPP in which every set $P_i$ selects exactly $k$ vectors.

**Questions:**

- Can the theoretical result extend to other settings of DPP, e.g., different $P_i$ selects a different number of vectors, or certain constrained DPP?
- The observation of local optimality for greedy is interesting, does it have other applications?

**Limitations:**

No. I suggest discussing some social impact, e.g., whether the greedy is better or worse for fairness issues than local search.

---

> ### Author Rebuttal · Authors · 2023-08-06
>
> We thank you for your positive feedback. Below, we have addressed your comments and questions.
>
> > The paper is a follow-up work of [IMGR20]. The improved analysis for the greedy algorithm is interesting but not the first result of a composable coreset.
>
> This is correct. However, our algorithm closes this line of work ([MIGR'19], [IMGR20], and [MKSK13]) by showing that the practical greedy algorithm provides an almost optimal approximation guarantee. We note that the motivation of [MIGR19] for studying local search for composable coresets is because simplicity is desirable to ensure that the algorithm is practical. Greedy avoids a lot of computation that is required for local search with a comparable guarantee. Furthermore, this paper provides a very different way of analyzing the relationship between local search and the greedy algorithm in comparison to [IMGR20] and could be interesting in its own right.
>
> -----
> > The studied problem is limited, e.g., only consider unconstrained DPP in which every set $P_i$ selects exactly $k$ vectors.
>
> > Can the theoretical result extend to other settings of DPP, e.g., different $P_i$ selects a different number of vectors, or certain constrained DPP?
>
> In this paper, similar to [MIGR19] and [IMGR20], we consider the basic setting of unconstrained determinant maximization. The goal is to get as small of a coreset as possible for each pointset $P_i$ regardless of its initial size $|P_i|$. We want to emphasize that for this problem, a coreset of **size $k$ is always necessary** as otherwise we cannot get any approximation guarantee. (To see this consider the setting where $P_i$ contains $k$ points with very high volume but the rest of the point sets $P_j$ have all points equal to 0. In this case the coreset size should be at least k).
>
> It is an interesting direction for future work to see if the greedy algorithm can be used to get a coreset for this problem under fairness constraint.
>
> ---------
>
> > The observation of local optimality for greedy is interesting, does it have other applications?
>
> The locality of greedy does have a consequence: the locality property directly recovers the k! guarantee of volume maximization in the offline case [CMI09], and marginally improves the guarantee to (k!)^(0.5 + o(1)). This is a different proof from the original paper and may be of interest independent of our composable coresets result. The statement and proof of this result can be found in the global response.
>
> ------------
> > I suggest discussing some social impact, e.g., whether the greedy is better or worse for fairness issues than local search.
>
> While our experiments focus on computing the empirical *local optimality* of the greedy algorithm, we expect that neither of the greedy nor local search will provide a fair result (e.g. consider the scenario where there are two populations in the data set where one population contains points with much higher volume. In this setting, both the greedy and local search prefer that population and are not fair in that respect).
> In order to get fair results one needs to change the formulation of this problem which is not the focus of this work.

---

### Official Review · Reviewer_vec7 · 2023-07-06

**Soundness:** 3 good
**Presentation:** 3 good
**Contribution:** 3 good
**Rating:** 6
**Confidence:** 4

**Summary:**

This work studies the determinant maximization problem, in which the input is a set of $n$ vectors in $d$ dimensions, and the goal is to select a subset of $k\leq d$ of these vectors which maximizes the determinant of the Gram matrix of the $k$ vectors, which is also the volume of the parallelepiped spanned by the vectors. This is an important classical problem with connections to diversity.

A popular heuristic for this problem is the greedy algorithm, which iteratively builds a set of vectors one at a time by selecting the vector that maximizes the improvement in the current volume. In prior work, an analysis of the greedy algorithm was given, which showed that it gives a $C^{k^2}$ factor approximation. However, the best known lower bound is $\Omega(k^{k-o(k)})$ (for composable coresets? not sure what the exact setting is here). This work shows that the analysis of the greedy algorithm can in fact be tightened to approximately match this upper bound, giving an approximation factor of $O(k)^{3k}$. This result is shown by proving that the greedy algorithm is approximately locally optimal, which in turn implies the approximation guarantee from results shown in prior work.

**Strengths:**

The analysis of greedy is an extremely important and old question (the prior bound is over 10 years old), and the authors provide a very nice improvement to this result, and achieves a tight approximation over some class of algorithms. The analysis is nontrivial to carry out.

**Weaknesses:**

While the result is very important, the techniques might be viewed as incremental over the analysis of local search given in prior work, and the paper doesn't give much intuition or discussion on why this result might have been missed in prior work.

**Questions:**

* I could not find the referenced lower bound of $\Omega(k^{k-o(k)})$ in prior work (I checked https://arxiv.org/abs/1907.03197 and https://arxiv.org/abs/1807.11648), I would appreciate a pointer to a specific theorem stating this. In particular, I would like to understand the exact setting in which this lower bound applies.

**Limitations:**

Yes

---

> ### Author Rebuttal · Authors · 2023-08-05
>
> We thank you for reviewing our paper and your positive feedback. In what follows we address your concerns and questions.
>
> ----------
>
> > While the result is very important, the techniques might be viewed as incremental over the analysis of local search given in prior work
>
> [IMGR20] discusses and analyzes the relationship between greedy and local search by utilizing a very lossy reduction to the k-perpendicular heights problem. In contrast, this paper provides a very different way of performing this reduction and is interesting in its own right.
>
> -------
>
> > I could not find the referenced lower bound of $\Omega(k^{k-o(k)})$ in prior work ... I would appreciate a pointer to a specific theorem stating this. In particular, I would like to understand the exact setting in which this lower bound applies.
>
> The lower bound is **Theorem 1.4** in the arxiv version of reference **[IMGR20]** “Composable Core-sets for Determinant Maximization Problems via Spectral Spanners.”
>
> The setting for our results is the **composable coreset** setting where the data is partitioned into multiple machines: machine $i$ holds the dataset $P_i$. The goal is to use greedy to summarize $P_i$, getting a subset $S_i=Greedy(P_i)$ such that $\mathrm{MAXVOL}^2_k(S) >= (1/\alpha) \mathrm{MAXVOL}^2_k(P)$ where $S=\sum_i S_i$, and $P = \sum_i P_i$, and where for a set $A$ its $\mathrm{MAXVOL}_k(A)$ is defined to be the maximum volume one can achieve by picking $k$ points in $A$. The lower bound shows that one cannot get $\alpha$ which is smaller than $k^{k(1-o(1))}$.
>
> Finally, in the offline setting we show that locality property one can marginally improve the k! guarantee of volume maximization of [CMI09], to $(k!)^{(0.5 + o(1))}$. Please refer to the global response for the proof.
>
> --------
> Please let us know if we can provide further clarification.

---

> > ### Comment · Reviewer_vec7 · 2023-08-10
> > **Thank you for the clarification on the lower bound.**
> >
> > I encourage the authors to include this reference in the main text, since it is crucial for understanding the claim of optimality.
> >
> > If you could give at least a partial answer to the question "why this result might have been missed in prior work", this would be very helpful. In particular, the main trick seems to be the use of the matrix determinant lemma, but this seems to be a common tool in the area of determinant maximization, so I find it somewhat surprising that this kind of observation was not made before. Would it be possible to give some more insight into what might have been the "key" that prior work didn't have, or maybe a particular difficulty in the calculations? It is of course possible that it was just missed, I'm just wondering if the authors had any take on this.

---

> > > ### Author Response · Authors · 2023-08-11
> > >
> > > We will make sure to include the reference in the final version.
> > >
> > > Regarding why this was missed previously, in short, those papers were pursuing different approaches as described below. Our goal however was to understand the local optimality property of the greedy algorithm and answering "does the greedy algorithm have similar properties as the local search algorithm?" While simple in retrospect, it is not clear why it should be helpful at the first look. Please see below for more details.
> > >
> > > In the composable coreset setting, in [MIGR19]: they used a more geometric based approach. In particular, the key idea was that a coreset for the directional height problem implies a composable coreset for determinant maximization. So it remained to show that a simple determinant maximization algorithm yields a coreset for directional height. The authors observed that local search was very helpful in proving the k-directional heights result. When the authors returned to the vanilla greedy algorithm, they tried to analogously perform the same reduction, but without the local optimality assumption. Without local optimality, the guarantee was much weaker.  Our key contribution is the "local optimality property" of the greedy algorithm. Therefore we can directly use the result of [MIGR19] on Local Search, to get our result for greedy, thus bypassing the lossy reduction which tries to get a coreset for directional height directly from greedy.
> > >
> > > In the composable coreset setting in [IMGR20]: they follow an approach based on spectral spanners. Their goal was to get the best constant in the exponent (e.g. k^{k/2}) and thus their algorithm is based on solving an LP and is not optimized for simplicity and practical use.
> > >
> > > In the offline setting: the reason it was probably missed is that swapping a vector into the greedy solution can break the "structure" of the greedy solution after even the first swap, so the vectors have to be carefully introduced in a sequence. So thinking of the structural property of local optimality would not be the first approach: indeed, [CMI09] iteratively compares the greedy and optimal solutions in a very different manner.

---

### Official Review · Reviewer_3pA1 · 2023-07-07

**Soundness:** 3 good
**Presentation:** 3 good
**Contribution:** 2 fair
**Rating:** 6
**Confidence:** 3

**Summary:**

The paper gives a tighter analysis for the greedy algorithm for constructing composable coreset for the determinant maximization problem. The greedy algorithm performs well in practice; however, the previous analysis was giving an approximation factor much farther from the lower bound. The authors in the paper bring the approximation factor for greedy algorithm much closer to the lower bound. this is achieved by showing that by just swapping a single point from the greedy solution with a vector obtained using local search approach does not increase the volume (determinant) by much. The authors also perform some experiments to show that in practice this local optimality property of greedy algorithm is actually much better than theoretical bounds.

**Strengths:**

The paper is well written and clear.
I tried to go through the proofs and unless I have missed something, the proofs are sound. Infact the proofs look quite elegant using techniques from simple linear algebra.
Code is provided for the empirical results.
By closing the gap between the lower bound and the approximation factor given by the greedy algorithm, the paper is able to give better explanation for the good performance of greedy algorithm in practice.

**Weaknesses:**

The only weakness I could think of is that the contribution is only theoretical and does not really have much practical implication as greedy algorithm was already known to perform nicely in practice. The simplicity of the proof techniques while not really a weakness may not excite the community much from the novelty perspective.



**Questions:**

Please refer to weaknesses.

**Limitations:**

Please refer to weaknesses.

---

> ### Author Rebuttal · Authors · 2023-08-05
>
> We thank you for reviewing our paper and your positive feedback. We argue that while this is primarily a theoretical contribution, it is valuable for the following reasons:
> - While it was already known that greedy does well in practice, our locality theorem, along with our experiments showing a lower value of local optimality on real datasets, provide a sound explanation for the strong practical performance of greedy in the context of composable coresets.
>
> - We further believe that the locality result on the greedy algorithm is a structural result, and may find uses in other contexts; we point the reviewer to the global response for an example on how this property recovers (and in fact slightly improves) the best analysis of Greedy in the offline setting.

---

> > ### Comment · Reviewer_3pA1 · 2023-08-17
> > **Replying to Rebuttal**
> >
> > Thanks for the response. I will keep my score

---

### Official Review · Reviewer_rYtN · 2023-07-07

**Soundness:** 3 good
**Presentation:** 2 fair
**Contribution:** 3 good
**Rating:** 6
**Confidence:** 3

**Summary:**

The paper considers the problem of picking k among n vectors that maximize the volume of the parallelepiped spanned by the selected vectors, which is equal to the squared determinant of the corresponding Gram matrix. In particular, the authors focus on the composable coreset setting of the problem where given a large dataset split into multiple subsets, the aim is to find small summaries (coreset) of each subset such that the union of the summaries is a good summary of the full dataset. Existing work showed that the best approximation possible for this problem is $\Omega(k)^{k - o(k)}$ and that an LP-based algorithm achieves almost optimal $\tilde{O}(k)^k$-approximation. Another Greedy algorithm followed by local search, which works better in practice than the LP-based one, was also shown to achieve ${O}(k)^{2k}$-approximation, while the Greedy algorithm alone is only shown to achieve $C^{k^2}$ for some constant $C$. This paper provides an improved approximation guarantee of ${O}(k)^{3k}$ for the Greedy algorithm, by providing an almost tight bound on the local optimality of the solution outputted by Greedy. The presented numerical results also show that the local optimality of the Greedy solution is significantly lower than the theoretical upper bound on real and random datasets.


**Strengths:**

- The paper provides a new analysis to the greedy algorithm which significantly improves upon its known approximation guarantee. Even though this approximation is still worse than the guarantees achieved by the LP-based and the Greedy + local search algorithms, this result is interesting as in practice the greedy algorithm performs better than the LP-based one, and skipping the local search stage saves time.

- The theoretical results are correct and experimental results illustrate nicely that the local optimality of the Greedy solution in practice is significantly better than the theoretical bound.


**Weaknesses:**

- The paper is not well self-contained; several details are omitted in the discussion and proofs which affect clarity (see questions for details).


**Questions:**

Suggestions to improve clarity:
- The Greedy algorithm description in the preliminaries Section 1.1.1 does not match the pseudocode in Algorithm 1 with no mention of the relation between them. It is good to explicitly discuss the relation between the two, even if this is possibly well known.
- In the proof of Theorem 5, the vectors in V are implicitly assumed to be linearly independent. This should be stated explicitly, explaining why this can be assumed without loss of generality.
- Provide details for the expression of $\mathrm{vol}(V)$ given in Theorem 5 and for why $|a^j_l| \leq 1$, or at least a clear reference. Similarly for other steps in this proof and other ones.
- Provide a reference for Lemma 6.

- In the experiments, do you use the data points directly as feature vectors, or you apply a kernel as in prior work?

Minor comments/suggestions:
- using $\mathrm{vol}(S)$ to denote the square volume is a bit confusing. I propose to use $\mathrm{vol}^2(S)$.
- typo line 87: $(2 k ( 1 + \epsilon))^k \rightarrow (2 k ( 1 + \epsilon))^{2k}$
- typo line 154: $a^j_i \rightarrow |a^j_i|$

**Limitations:**

yes

---

> ### Author Rebuttal · Authors · 2023-08-05
>
> To Reviewer rYtN,
>
> We thank you for your positive feedback. Below, we have addressed your comments and questions.
>
> -----
>
> > The Greedy algorithm description in the preliminaries Section 1.1.1 does not match the pseudocode in Algorithm 1 with no mention of the relation between them. It is good to explicitly discuss the relation between the two, even if this is possibly well known.
>
> Thanks, we will add clarification about why picking the vector with the largest perpendicular distance to the current solution is the same as greedily picking the vector that maximizes the volume.
>
> -----
>
> >In the proof of Theorem 5, the vectors in V are implicitly assumed to be linearly independent. This should be stated explicitly, explaining why this can be assumed without loss of generality.
>
> If the greedy solution is not linearly independent, then that means the rank of the n vectors is less than k, so both the greedy and optimal volume is 0. In this case, Theorem 5 directly holds without proof. We will add this explicitly as suggested.
>
> ----
>
> >Provide details for the expression of $\mathrm{vol}(V)$ given in Theorem 5 and for why $|a^j_l| \leq 1$, or at least a clear reference. Similarly for other steps in this proof and other ones.
>
> * The main idea behind the volume expansion is that subtracting a vector's perpendicular components from other vectors does not change the volume, similar to the Gram-Schmidt volume computation. We will elaborate on this further in the final version.
>
> * |alpha_l^j| <= 1 directly holds by the property of the greedy algorithm: if |alpha_l^j| > 1, then v_j would have been chosen before v_i. We will add details about this.
>
> * We will ensure that appropriate explanations are added for any ambiguous steps.
>
> ----
>
> > Provide a reference for Lemma 6.
>
> We apologize for missing this reference. A proof for this lemma can be found in the paper titled _Eigenvalues of rank-one updated matrices with some applications_, by Jiu Ding and Aihui Zhou. We will include this reference in the final version of our paper.
>
> -----
>
> >In the experiments, do you use the data points directly as feature vectors, or you apply a kernel as in prior work?
>
> We use the data points directly as feature vectors. However, we don't expect the results to vary much. As an example, we applied the RBF kernel with sigma = 6 as used in prior work and repeated Experiment 1 on the GENES dataset for k = 2,4,...,20 and the random dataset. The local optimality was still very close to 1 (figure attached in the global response).
>
> -----
>
> > using $\mathrm{vol}(S)$ to denote the square volume is a bit confusing. I propose to use $\mathrm{vol}^2(S)$.
>
> Our analysis focuses on $\mathrm{vol}(S)$, while $\mathrm{vol}^2(S)$ is the determinant. For $\mathrm{vol}^2(S)=$ determinant, we obtain a guarantee of $k^{3k}$, while local search and the optimal algorithm are known to have guarantees of $k^{2k}$ and $k^k$ respectively. If we instead want to write the guarantees in terms of volume, they become $k^{1.5k}$, $k^k$ and $k^{k/2}$ for greedy, local search and optimal respectively. We will make this distinction more clear in the final version.
>
> ------
>
> > typo line 87: $(2 k ( 1 + \epsilon))^k \rightarrow (2 k ( 1 + \epsilon))^{2k}$
>
> > typo line 154: $a^j_i \rightarrow |a^j_i|$
>
> We apologize for the mistakes in the submission. We will correct all typos in the final version.

---

> > ### Comment · Reviewer_rYtN · 2023-08-11
> > **Response to rebuttal**
> >
> > Thank you for your response and additional experiment. The additional derivation of the $k!$ guarantee of offline greedy is also interesting.
> > I highly recommend including more detailed explanations of the proof steps, beyond what you stated in the rebuttal. This would increase the readability of the paper to a wider audience.

---

> > > ### Author Response · Authors · 2023-08-11
> > >
> > > Thank you for finding this application interesting and for your comment. We will certainly describe the proof steps in further detail for the final version.

---

### Author Rebuttal · Authors · 2023-08-06

We thank all the reviewers for their positive and constructive feedback.

Bellow, we give a proof that the locality property can recover and in fact marginally improve the k! guarantee for the offline version of the greedy algorithm [CMI09]. We will include it in the final version. While the focus of our paper is on composable coresets, we hope that this application in the offline case demonstrates why our structural result on local optimality could be useful in analyzing the greedy algorithms in other contexts as well.

------

### Theorem
Let $P$ be a point set, Greedy$(P) = \\{v_1,\ldots,v_k\\}$ the output of the greedy algorithm and $\text{maxvol}_k(P)$ the maximum volume of any subset of $k$ vectors from $P$. Then

$\text{vol}(\text{greedy}(P)) \geq \frac{maxvol_{k}(P)}{\prod_{i=2}^k (1+\sqrt{i})}$.

### Proof
Let $S \subseteq P$ be the set of $k$ vectors with maximum volume. Without loss of generality and for simplicity of exposition, we assume $S\cap \text{Greedy}(P) = \varnothing$ (the proof still goes through if this is not the case).

Consider the set $W_1 = \\{v_1\\} \cup S$ with $k+1$ elements. Perform the greedy algorithm on $W_1$ with $k$ steps. Clearly, greedy will choose $v_1$ first and then some $k-1$ of the remaining vectors. Label the left out vector $w_1$.

Inductively define $W_{i+1} = \\{v_1,\ldots,v_i,v_{i+1}\\} \cup (S - \\{w_1,\ldots,w_i\\})$, which has size $k+1$. Perform greedy on $W_{i+1}$ with $k$ steps. The first $i+1$ vectors chosen will be $v_1,\ldots,v_i,v_{i+1}$ by definition. Call the left out vector $w_{i+1}$. We now have an ordering for $S = \\{w_1,\ldots,w_k\\}$.

Starting with the greedy solution, we will now perform $k$ swaps to obtain the optimal solution. Each swap will increase the volume by a factor of at most $1+\sqrt{k}$. Initially, our solution starts with Greedy$(P) = \\{v_1,\ldots,v_k\\}$. Note that this is also the output of greedy when applied to the set $\text{Greedy}(P) \cup \\{w_k\\} = W_k$. Swapping in $w_k$ in place of $v_k$ increases our volume by a factor of at most $1+\sqrt{k}$.

Our current set of vectors is now $\\{v_1,\ldots,v_{k-1},w_k\\}$. By the ordering on $S$, this is also the greedy output on the set $W_{k-1} = \\{v_1,\ldots,v_{k-1},w_{k-1},w_k\\}$. Therefore, we may swap in $w_{k-1}$ in place of $v_{k-1}$ in our current set of vectors by increasing the volume by at most a factor of $(1+\sqrt{k})$. Proceeding in this manner, we can perform $k$ swaps to obtain the optimal solution from the greedy solution by increasing our volume by a factor of at most $(1+\sqrt{k})^k$.

To obtain the slightly better approximation factor in the theorem statement, we observe that in the proof of theorem 5 in the paper, swapping out the $i^{\text{th}}$ vector from the greedy solution for a vector that was not chosen increases the volume only by a factor of $(1+\sqrt{k+1-i}) \leq 1 + \sqrt{k}$, and that swapping out the $k^{\text{th}}$ vector does not increase the volume at all. Therefore, the approximation factor of greedy is at most

$\prod_{i=1}^{k-1} (1+\sqrt{k+1-i}) = \prod_{i=2}^k (1+\sqrt{i})$.

### Remark
Note that $\prod_{i=2}^k (1+\sqrt{i}) < 2^k \sqrt{k!}$ for $k \geq 7$, which is $(k!)^{\frac{1}{2} + o(1)}$. While the improvement in the approximation factor is quite small, we emphasize that the proof idea is very different from the $k!$ guarantee obtained in [CMI09].

----------

Description of attached figure: We repeated Experiment 1 for the GENES dataset with the RBF kernel applied as requested in Review 1.

---

> ### Author Response · Authors · 2023-08-16
>
> We hope that our responses have clarified all questions about our paper, and we will make sure to incorporate your feedback in our final version. If you believe that all major concerns have been addressed, could you please consider increasing the scores to support the inclusion of our paper in NeurIPS? Thank you so much!

---

### Decision · Program_Chairs · 2023-09-21

**Decision:**

Accept (poster)

**Comment:**

The reviewers are generally supportive of the paper. The reviewers appreciate the improved guarantee for the practical greedy algorithm, now qualitatively similar to the best known bound for all algorithms. On the other hand, some reviewers note that the techniques are not groundbreaking and could be perceived as incremental from the local search analysis. Nevertheless, several reviewers think the technique could be of independent interest, beyond the specific task considered.

The authors are advised to improve the writing, especially by including the intuition and the connection between the new technique and other related problems from the discussion with the reviewers.